# Phosphorylated Fish Gelatin and the Quality of Jelly Gels: Gelling and Microbiomics Analysis

**DOI:** 10.3390/foods12193682

**Published:** 2023-10-07

**Authors:** Shiyu Wu, Wanyi Sun, Yihui Yang, Ru Jia, Shengnan Zhan, Changrong Ou, Tao Huang

**Affiliations:** 1College of Food and Pharmaceutical Science, Ningbo University, Ningbo 315211, China; wsyerbing@163.com (S.W.); 18071965076@163.com (W.S.); 17809392891@163.com (Y.Y.); jiaru@nbu.edu.cn (R.J.); zhanshengnan@nbu.edu.cn (S.Z.); ouchangrong@nbu.edu.cn (C.O.); 2Key Laboratory of Animal Protein Food Deep Processing Technology of Zhejiang Province, Ningbo University, Ningbo 315832, China

**Keywords:** phosphorylation, fish gelatin, jelly, quality variation, microorganism

## Abstract

Phosphorylated fish gelatin (PFG) exhibited preferable physical and chemical properties than fish gelatin (FG) in our previous study. To investigate the application values of PFG, the effects of different ratios (2:1, 1:1 and 1:2) of FG(PFG)/κ carrageenan (κC) on the quality of jelly gels (JGs) were investigated. The sensory quality of PFG:κC (1:2)/FG:κC (1:2) was found to be superior based on sensory evaluations, which was also verified with the results for texture, rheology, etc. Moreover, the structural changes in JGs were related to the introduction of phosphoric acid groups into the molecular chain of gelatin and the protein–polysaccharide interactions. According to the storage results, PFG jelly had better storage quality, higher hardness and chewiness values than those of FG jelly. High-throughput sequencing of JG microbial analysis showed that the addition of PFG changed the amount of microorganisms, microbial species abundance and the microbial composition of JGs, which were also closely related to the storage quality of JGs. In conclusion, the applications of PFG have promising potential to improve the quality of confectionery.

## 1. Introduction

Jelly is a popular leisure food owing to its good taste and swallowability. The normal procedure with jelly mainly includes heating, injection, molding and shaping [1]. Generally, agar has been widely used as a gelling agent to increase the gelling strength of jelly. Moreover, carbohydrates, thickening agents and food additives (such as sweetening agents, colorants, souring agents, etc.) are added to improve the texture and sensory properties of jelly manufacturing. At present, sugar-based carbohydrates are still the main ingredients in jelly processing. For example, in the production of guava–pineapple jelly, the amount of sugar reaches 42.66% (*w*/*w*) [2]. In general, the sugar sources in jelly products are mainly glucose, saccharose, fructose syrup, etc., which may lead to a series of severe health problems, such as fatty liver, dental caries and obesity [3]. Clearly, jelly produced in that way is not suitable for diabetics. Therefore, it is urgent to develop sugar-free and low-calorie jelly products to meet the needs of modern people [4].

Gelatin has been successfully used to enrich the flavor, stability, chewability and nutrients of dairy products, baked foods, candies and meat products [5]. Gelatin is a good source of protein and possesses excellent gelling properties which could endow the products with the chewable enjoyment of elasticity. Moreover, gelatin is one of the few hydrocolloids that does not cause a rise in blood sugar like carbohydrates. Therefore, gelatin can be a superior alternative to carbohydrates as a gelling agent in jelly manufacturing. Statistically, approximately 98.5% of global commercial gelatin is derived from pig skin, cow skin and bone [6]. However, the consumers constituting the target market of jelly products include consumers with regional and socio-cultural differences [7]. For example, Muslims and Jews refuse to eat foods that contain pork because of their religious beliefs. Indians eat neither pig- nor cow-based products. Gelatin products derived from mammals, such as pigs or cattle, also carry the risks of foot-and-mouth disease and mad cow disease [8]. Fish gelatin (FG) has similar physical and chemical properties to mammalian gelatin without any of its drawbacks. However, the contents of proline and hydroxy proline in FG are low [9]. Thus, FG does not have enough hydrogen bonds between the hydroxyl group and water to form the stable, three-strand helical structure of gelatin gel, showing poor gelling properties [10]. This drawback limits its application in the food industry.

Phosphorylation has been proved to be an important and effective method to improve the functional characteristics and stability of food proteins [11]. The introduction of phosphate groups could enhance the electro-negativity of protein molecules, improving the electrostatic repulsion between protein molecules and promoting the functional properties of protein, such as solubility, foaming and emulsification [12]. Previously, we successfully prepared phosphorylated FG with higher gelling properties than the unmodified one. The κC, a high-molecular-weight polysaccharide, exhibits remarkable gelling thickening and is extensively employed as a food thickener, suspension agent, emulsifier and stabilizer in the food industry [13]. However, there is still a lack of basic information focusing on modified gelatin and κC applied to jelly products. Therefore, this study aimed to develop a new type of healthy jelly product to solve the problem of the high sugar content in available jelly products and investigated the effect of phosphorylated fish gelatin (PFG) on the quality of jelly from the perspectives of a sensory test, a textural analysis and a rheological analysis; the relationship between quality change in jelly and microorganisms was also addressed. We hope that this paper can provide some useful thoughts on the production of high-quality jelly products.

## 2. Material and Methods

### 2.1. Chemicals and Reagents

Fish gelatin (FG, B type) was purchased from Shanghai Yuanye Biotechnology Co., Ltd. (Shanghai, China). Sodium pyrophosphate was purchased from Shanghai MacLean Biotechnology Co., Ltd. (Shanghai, China). Ziplock bags (polyethylene) were purchased from Tianjin Anhua Plastics Co., Ltd. (Tianjin, China). The κ-carrageenan (κC), maltitol, green tea essence, potassium sorbate and sodium citrate were of food analysis grade. They were purchased from Sigma-Aldrich (Diegem, Belgium). The other reagents were of analytical grade.

### 2.2. Preparation of Fish Gelatin Jelly Candy

The phosphorylated fish gelatin (PFG) was prepared according to our previous reports [14]. The gelatin–κC mixture solution was prepared by dissolving different ratios of PFG or FG to κC, namely 2:1, 1:1 and 1:2, in distilled water. The mixture solution was heated at 80 °C with 100 rpm constant stirring for complete dissolution. Then, 0.1% (*w*/*w*) sodium citrate, 0.1% (*w*/*w*) green tea essence, 0.1% (*w*/*w*) potassium sorbate, 18% (*w*/*w*) maltitol and 16% (*w*/*w*) skim milk powder were added to the gelatin–κC mixture solution successively and then heated together at 120 °C for 30 min. Then, the mixture was transferred to a mold to form jelly gels (JGs) at room temperature. The JGs were sealed and stored at 4 °C for the next analysis. The prepared JGs with unphosphorylated FG were set as the control group. In addition, the prepared JGs using unphosphorylated FG and κC at ratios of 2:1, 1:1 and 1:2 were named FG:κC (2:1), FG:κC (1:1) and FG:κC (1:2), respectively. The prepared JGs using PFG and κC at ratios of 2:1, 1:1 and 1:2 were named PFG:κC (2:1), PFG:κC (1:1) and PFG:κC (1:2), respectively. The specific composition of the sample is shown in Appendix A.

### 2.3. Sensory Evaluation

The sensory scores of all JGs were evaluated according to the sensory scoring standard in China (GB 19883-2005) [15] with some appropriate modifications. Ten volunteers who passed the sensory index training evaluated the JGs from the perspectives of taste, textural acceptance, color, hardness and elasticity (Appendix A).

### 2.4. Rheological Properties

The rheological properties of all JGs were determined using the rheometer (Discovery Hybrid Rheometer, TA Instrument, New Castle, DE, USA). Strain evaluation was measured as strain ranging from 0.1% to 100% using a 20 mm diameter probe. The gap value was set as 1000 μm; the frequent cohesive energy (Ec) was calculated using the following equation [16]:
Ec = 0.5γ^2^c_r_G′,
where c_r_ is the critical strain.

### 2.5. Storage Quality of JGs

#### 2.5.1. Textural Properties

The textural properties of JGs (d 2 cm × h 2 cm) were evaluated using a texture analyzer (TA.XT.Plus, Stable Micro System, Surrey, UK) equipped with a P 36R probe. The measurement speed was 1 mm/s; the deformation was 40%. Each sample was measured in 5 parallels, and the hardness and elasticity parameters were recorded at the same time [17].

#### 2.5.2. Color Analysis

The color change of JGs was measured using a portable colorimeter (CR-400, Konica Minolta, Inc., Tokyo, Japan). Before the experiment, the colorimeter had been calibrated. The brightness (L*), red–green deviation (a*) and yellow–blue deviation (b*) were recorded. The whiteness value was calculated using the equation below. Each sample was performed in triplicate [18].


$$W\;=\;100\;-\;\sqrt{{\left(100-L*\right)}^{2}+{a*}^{2}+{b*}^{2}}$$


#### 2.5.3. Deformation Resistance

The JGs were sealed in a vacuum bag and then transferred to a constant temperature and humidity incubator at 37 °C for 24 h. The height change in the cross sectional area (CAS) in the process of jelly insulation was measured [19]. The deformation resistance was calculated according to the following equation. Each sample was performed in triplicate.
Deformation resistance = H_2_/H_1_ × 100%,
where H_1_ refers to the height before heat preservation, and H_2_ is the height after heat preservation.

#### 2.5.4. Microbial Diversity

The FG–JGs3/PFG–JGs3 samples had the highest sensory scores and the best physical–chemical properties. For the following experiment, we mainly evaluated microbial diversity changes of FG:κC (1:2) and PFG:κC (1:2) during 14-day storage. About 1g of JGs was pre-treated with phosphate buffer, and the genomic DNA of microorganisms in the JGs was extracted using an E.Z.N.ATMMag-Bind Soil DNA Kit. The genomic DNA was used as a template. PCR amplifications were performed twice on the bacterial 16srDNA V3–V4 region and the fungal ITS1–ITS2 region gene sequences, respectively. After the amplification, the library was detected using 1.5% agarose gel electrophoresis, and the concentration of the library was determined using a Qubit 3.0 fluorescence quantitative analyzer. Then, the constructed amplicon library was sequenced with PE250 using the Illumina Nova 6000 platform.

### 2.6. Statistical Analysis

Data were expressed as the means ± standard deviation (SD). Statistical analysis of the results was performed using SPSS 23.0 software. A one-way analysis was used to evaluate significant differences (*p* < 0.05) between groups. Origin 2021 software was used for generating pictures.

## 3. Results and Discussion

### 3.1. Sensory Evaluation

Sensory evaluation is a crucial way to evaluate the acceptance of products [7]. As depicted in Figure 1, the FG:κC (1:2) and PFG:κC (1:2) possess the highest taste, texture, hardness and elasticity values and sensory scores when compared to those of other JGs. This may be due to the formation of more stable FG–κC complexes through electrostatic forces and hydrogen bonds, which enhance the viscosity and the hardness of JGs [20]. As the proportion of κC increased, the gel strength of the jelly slightly increased, which may have given the jelly better elasticity and hardness and higher sensory scores. Similar results were also found in Cheng’s study [21]. Meanwhile, at the same ratio of FG/κC, the sensory scores of samples had non-significant differences (*p* > 0.05). This indicated that the sensory panel volunteers could not perceive the difference between the PFG–JGs and FG–JGs. Based on the results of sensory scores, only FG:κC (1:2) and PFG:κC (1:2) were used for the following storage experiments.

### 3.2. Analysis of Jelly Deformation and Fracture Properties

Deformation and fracture properties are the key indicators for evaluating gelatin products [16]. As shown in Figure 2, the storage moduli (G′) of all JGs were higher than those of the loss moduli (G″), as the strain was 0.1%. However, as the strain increased to ranges of 10–100%, the G′ was significantly lower than the G″, indicating that the JGs had experienced significant deformation and fracture. During the deformation process, the G′ and G″ had an intersection point which was usually defined as critical strain (Cr). The changes in Cr, intersection moduli and cohesive energy (Ec) of all JGs are shown in Table 1. It shows that with the increase in FG/PFG, the Cr increased and then decreased, the intersection modulus decreased and then increased, while the Ec decreased. In general, electrostatic interactions, especially the interactions of hydrogen bonding and hydrophobia may lead to associative interactions between gelatin and κC [20]. Compared with FG:κC (2:1) and PFG:κC (2:1), the interaction between an FG and a κC of FG:κC (1:2) and PFG:κC (1:2) may be enhanced, while the Cr, intersection modulus and Ec of the JGs were decreased. Moreover, the introduced phosphate groups might alter the structure of the original FG, thus improving the gelling properties. Therefore, at the same ratio, PFG could maintain the structure of the JGs with the higher Cr, intersection modulus and Ec.

### 3.3. Storage Quality of JGs

#### 3.3.1. Textural Properties

Texture can be used to evaluate the effect of the tongue and teeth on the gel, and the higher the textural properties’ parameters (hardness, cohesion, chewiness, resilience and adhesion), the better the quality of gels [22]. As shown in Table 2, with the increase in κC, the JGs present better textural properties, which may be attributed to the interaction of FG and κC through electrostatic interaction. The combination of gelatin and agar resulted in the formation of gels with a firmer gel texture, which was possibly due to the protein–polysaccharide interactions [23]. Moreover, skimmed milk protein also could interact with κC through hydrogen bonds or electrostatic interaction to form a stable gel network that supported the jellies’ gel network [24]. At the same ratios of FG/κC, the textural properties of the PFG–JGs were better than those of the FG–JGs. This may be related to the introduction of phosphoric acid groups into the molecular chain of gelatin, which can enhance the ionic interaction between the phosphate groups in gelatin and the NH^3+^ of amino acids, leading to the aggregation of proteins [25]. The chewiness of FG:κC (2:1), FG:κC (1:1), FG:κC (1:2), PFG:κC (2:1), PFG:κC (1:1) and PFG:κC (1:2) was 5.22 ± 0.39, 0.45 ± 0.03, 6.02 ± 0.35, 5.75 ± 0.27, 2.36 ± 0.43 and 6.84 ± 0.45, respectively. The PFG–JGs showed a higher chewiness; this was because the addition of PFG resulted in a stronger three-dimensional gel network structure for JGs. With the prolongation of storage time, the hardness and chewiness of all samples increased first and then decreased, while there was no obvious change in elasticity and resilience. Similarly, the white tea candy showed an increase in hardness (from 1.4 to 5.1 g) during long-term storage due to syneresis [17]. However, with the extension of storage time, the JGs began to rot, deteriorate and metamorphize due to the oxygen and the increase in the amount of microorganisms and microbial abundance. Compared with FG jelly, PFG jelly possessed better hardness and chewiness, which indicated that the advantages of phosphorylation modification persisted after storage. A stronger network structure will reduce the water loss in storage and better maintain the internal gel structure, which may be related to hydrogen bonding, hydrophobia, etc. [26].

#### 3.3.2. Color Analysis

Color is closely related to consumer preferences and the acceptance of food products [27]. As shown in Table 3, FG:κC (1:1) shows the highest brightness (L*) value, while FG:κC (1:2) and PFG:κC (1:2) have the lowest L* values. These indicates that a high proportion of FG/PFG or κC could reduce the L* value of JGs. Among all samples, FG:κC (1:2) and PFG:κC (1:2) had the highest b* values. This may be because the κC (the inherent color of κC is yellowish) occupies a large proportion in the jelly which decreased the L* values. During storage, PFG–JGs had higher L*, a* and b* values than FG–JGs. This phenomenon was also found in the color change in Acacia (*Annona muricata* L.) jelly candy, which varied from bright yellow to dark yellow largely due to oxidation [28]. Therefore, it can be inferred that the oxidation degree of PFG–JGs was lower than that of the FG–JG group and resulted in fewer quality changes during storage.

#### 3.3.3. Analysis of Deformation Resistance Rate

Deformation resistance refers to the thermal stability of samples [29].The deformation resistance rates of all JGs are shown in Table 4. On day 0, FG:κC (1:1) and PFG:κC (1:1) showed the lowest resistance to deformation, as compared to the other groups, which was consistent with the lowest textural properties. The fluctuation of the deformation rate may be due to the associative interaction of FG and κC through electrostatic interactions and to a lesser extent to the hydrogen bonding and the hydrophobic interactions with the increase in the κC ratio [20]. FG:κC (1:2) and PFG:κC (1:2) indicated higher resistance to deformation and better textural properties. Moreover, when compared to FG–JGs, the deformation resistance values of the PFG–JGs were higher (at the same FG/κC ratio). This may be attributed to the various water contents, compositions, hardness and other textural properties. The deformation resistance values of all JGs decreased significantly and there was no significant difference between the PFG–JGs and the FG–JGs during storage. Generally, the change in textural properties of FG was affected by the concentration, temperature, pH and salt [30]. With the prolongation of storage time, the JGs possessed higher water content, which may lead to corruption and deterioration; the samples also became soft, showing a lower deformation resistance.

#### 3.3.4. Statistical and Quality Analysis of High-Throughput Sequencing

The growth and reproduction of microorganisms is the main factor affecting food quality. As shown in Table 5, the original sequence numbers of bacteria in all JGs are higher than that of fungi. The OUTs produced by the bacterial and fungal flora of FG:κC (1:2) 0 d, PFG:κC (1:2) 0 d, FG:κC (1:2) 7 d, PFG:κC (1:2) 7 d, FG:κC (1:2) 14 d and PFG:κC (1:2) 14 d samples were 387, 308, 329, 124, 126, 137 and 132, and 104, 92, 110, 121 and 97, respectively. The OUTs of the FG–JGs were higher than those of the PFG–JGs during storage. This indicated that the PFG–JGs had fewer microorganism species than the FG-JGs; the addition of PFG might improve storage quality by reducing microorganism species.

#### 3.3.5. Alpha Diversity Index Analysis of JGs

Alpha (α) diversity is extensively used to reflect the diversity and richness of microorganisms [31]. The α diversity-related indexes include the richness index, the Chao1 index and the reads index of the bacterial and fungal flora of all the JGs. As shown in Table 6, the Chao1 index is widely used to evaluate the abundance of microbial communities in JGs, and the higher Chao1 index value indicates a higher abundance of microbial communities. The Simpson index was negatively correlated with the structure of microbial flora, and the larger the Simpson index value, the simpler the structure of the microbial flora. The richness index value and the Chao1 index value of FG–JGs were higher than that of the PFG–JGs; thus, the PF–JGs possessed a lower abundance of microbial communities as compared with the FG–JGs. The Simpson index values of PFG:κC (1:2) 7d and PFG:κC (1:2) 14 d were higher than FG:κC (1:2) 7 d and FG:κC (1:2) 14 d, indicating that the structure of the microbial flora of the PFG–JGs was simpler than that of the FG–JGs during the storage process. It was speculated that the better storage quality of PFG–JGs related to a lower abundance of microbial species and a simpler structure of microbial flora.

#### 3.3.6. Analysis of Bacterial Flora in JGs at the Genus Level

Figure 3 shows the genus-level-based bacterial community structure of the JGs and the shared or unique flora. In all samples, there were 57 common bacterial groups, while FG:κC (1:2) 0 d, PFG:κC (1:2) 0 d, FG:κC (1:2) 7 d, PFG:κC (1:2) 7 d, FG:κC (1:2) 14 d and PFG:κC (1:2) 14 d had 122, 57, 69, 9, 10 and 8 unique bacterial groups, respectively (Figure 3b). After the same storage time, FG–JGs possessed more unique bacterial groups than PFG–JGs. The content of bacteria in the PFG–JG jelly was lower than that of the FG–JG except for *Pseudomonas* (Table 7). In addition, the content of harmful bacteria (such as *Shewanella*, *Acinetobacter* and *Carnobacterium*) in the PFG jelly was less than that of the FG jelly. Hence, the PFG jelly had better health benefits and storage quality.

#### 3.3.7. Analysis of Fungal Flora in JGs at the Genus Level

Figure 4 shows the genus-level-based fungal microbiota structure of JGs and the shared or unique microbiota changes. In all JGs, there were 32 common fungal groups, while FG:κC (1:2) 0 d, PFG:κC (1:2) 0 d, FG:κC (1:2) 7 d, PFG:κC (1:2) 7 d, FG:κC (1:2) 14 d and PFG:κC (1:2) 14 d had 19, 14, 7, 15, 10 and 7 unique fungal groups, respectively (Figure 4b). After the same storage time, the FG–JGs possessed more unique fungal groups than the PFG–JGs. The dominant fungi in JGs were the fungi that were not assigned to the genus level and were unidentified (Table 8). The amount of dominant fungi in the PFG–JGs was less than that in the FG–JGs; the amount of other identified fungi was not clear during storage.

## 4. Conclusions

In conclusion, FG and PFG were used to develop jelly products with different FG/κC ratios. FG:κC (1:2)/PFG:κC (1:2) had the best sensory score, texture characteristics and deformation resistance of the jelly. FG:κC (2:1)/PFG:κC (2:1) had the worst sensory score and other related indexes of the jelly. Compared to the FG–JGs, the PFG–JGs had better texture, color and resistance deformation properties during storage. The results of high-throughput sequencing indicated that the PFG–JGs had fewer microorganisms, abundance of microbial species and simpler microbial composition, which was also verified with the results of the analysis of bacterial/fungal flora in JGs at the genus level. The addition of PFG had a great impact on the storage quality of FG jelly.

## Figures and Tables

**Figure 1 foods-12-03682-f001:**
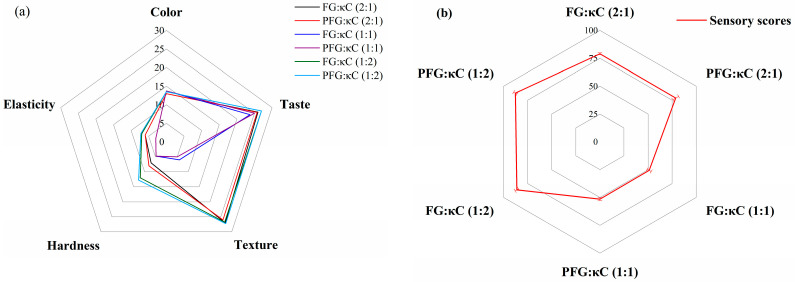
The difference in the taste, texture, color, hardness, elasticity (**a**) and sensory scores (**b**) among different FG/carrageenan ratios in JGs.

**Figure 2 foods-12-03682-f002:**
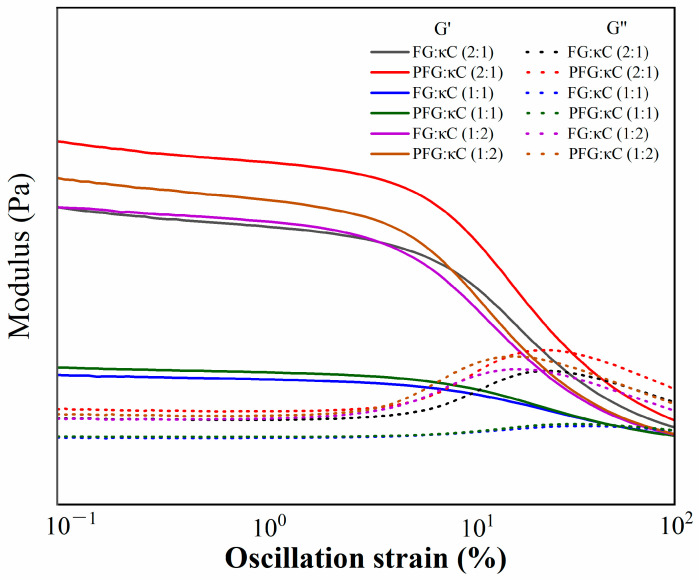
Variation in storage/loss modulus with strain for different JGs.

**Figure 3 foods-12-03682-f003:**
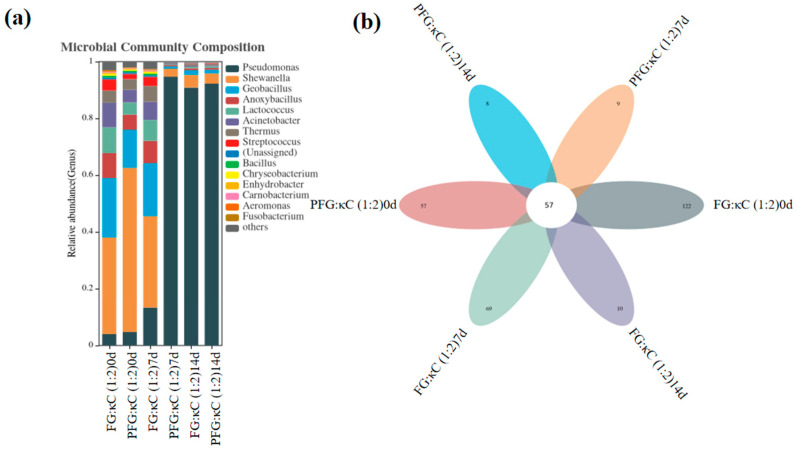
Variation in bacterial community structure (**a**), common or unique flora (**b**) Veen plot based on genus-level JGs.

**Figure 4 foods-12-03682-f004:**
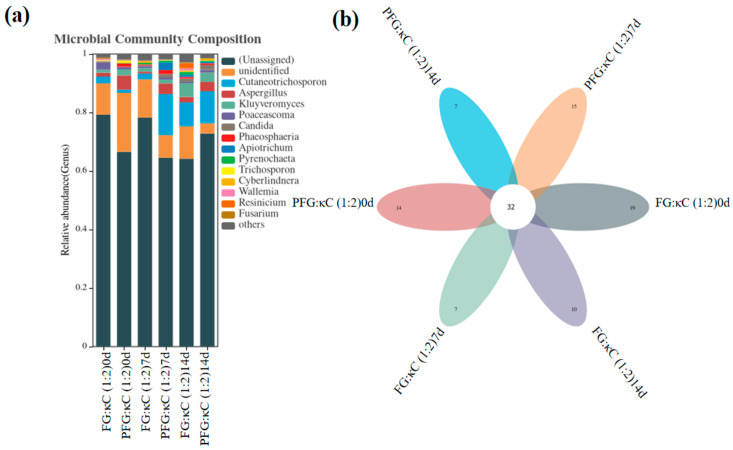
Variation in fungal community structure (**a**), common or unique flora (**b**) Veen plot based on genus-level JGs.

**Table 1 foods-12-03682-t001:** Changes in critical strain, intersection modulus and cohesive energy of JGs.

Sample Name	Critical Strain (Cr)	Modulus of Intersection	Cohesive Energy (Ec, J/m^3^)
FG:κC (2:1)	28.97 ± 4.82 ^b^	63.70 ± 6.45 ^b^	25,128.60 ± 435.27 ^b^
PFG:κC (2:1)	29.93 ± 0.61 ^b^	79.8 ± 2.57 ^a^	42,822.21 ± 754.03 ^a^
FG:κC (1:1)	52.29 ± 15.64 ^a^	16.55 ± 5.86 ^c^	24,458.40 ± 535.48 ^c^
PFG:κC (1:1)	49.84 ± 7.41 ^a^	17.63 ± 5.40 ^c^	22,306.51 ± 545.66 ^d^
FG:κC (1:2)	22.86 ± 6.90 ^b^	91.02 ± 11.63 ^a^	17,076.40 ± 492.93 ^e^
PFG:κC (1:2)	20.21 ± 4.20 ^b^	64.18 ± 12.12 ^b^	16,122.56 ± 582.25 ^f^

The values are the mean ± standard deviation, and the values with different letters in the same column are significantly different at *p* < 0.05.

**Table 2 foods-12-03682-t002:** Changes in textural properties of JGs during storage.

Storage Time/Day	Sample Name	Hardness (N)	Elasticity	Chewiness (N)	Resilience
0	FG:κC (2:1)	11.08 ± 0.63 ^d^	0.84 ± 0.02 ^cd^	5.22 ± 0.39 ^e^	0.27 ± 0.02 ^ab^
	PFG:κC (2:1)	11.64 ± 0.62 ^d^	0.86 ± 0.02 ^bc^	5.75 ± 0.27 ^de^	0.27 ± 0.01 ^abc^
	FG:κC (1:1)	2.89 ± 0.87 ^f^	0.71 ± 0.04 ^f^	0.45 ± 0.03 ^g^	0.11 ± 0.02 ^d^
	PFG:κC (1:1)	5.92 ± 0.57 ^e^	0.92 ± 0.00 ^a^	2.36 ± 0.43 ^f^	0.25 ± 0.02 ^c^
	FG:κC (1:2)	11.87 ± 0.56 ^d^	0.81 ± 0.01 ^de^	6.02 ± 0.35 ^d^	0.28 ± 0.00 ^ab^
	PFG:κC (1:2)	13.25 ± 0.78 ^c^	0.84 ± 0.02 ^cd^	6.84 ± 0.45 ^c^	0.27 ± 0.01 ^abc^
7	FG:κC (1:2)	15.06 ± 1.16 ^b^	0.81 ± 0.04 ^de^	7.66 ± 0.69 ^b^	0.28 ± 0.02 ^a^
	PFG:κC (1:2)	17.48 ± 1.03 ^a^	0.87 ± 0.01 ^b^	9.3 ± 0.58 ^a^	0.26 ± 0.01 ^abc^
14	FG:κC (1:2)	12.21 ± 0.32 ^cd^	0.79 ± 0.01 ^e^	6.02 ± 0.22 ^d^	0.26 ± 0.00 ^bc^
	PFG:κC (1:2)	15.11 ± 0.88 ^b^	0.86 ± 0.02 ^bc^	8.02 ± 0.81 ^b^	0.25 ± 0.01 ^bc^

The values are the mean ± standard deviation, and the values with different letters in the same column are significantly different at *p* < 0.05.

**Table 3 foods-12-03682-t003:** Changes in color of JGs during storage.

Storage Time/Day	Sample Name	L*	a*	b*	W
0	FG:κC (2:1)	55.47 ± 0.35 ^c^	−3.76 ± 0.04 ^g^	−4.10 ± 0.13 ^c^	55.13 ± 0.35 ^ab^
	PFG:κC (2:1)	54.33 ± 0.11 ^d^	−3.44 ± 0.04 ^f^	−3.69 ± 0.09 ^b^	54.05 ± 0.12 ^ab^
	FG:κC (1:1)	59.89 ± 0.22 ^a^	−4.14 ± 0.03 ^h^	−3.30 ± 0.02 ^a^	59.54 ± 0.22 ^abc^
	PFG:κC (1:1)	57.36 ± 0.15 ^b^	−3.68 ± 0.03 ^g^	−3.10 ± 0.08 ^a^	57.09 ± 0.15 ^a^
	FG:κC (1:2)	51.71 ± 0.49 ^e^	−3.24 ± 0.08 ^e^	−4.50 ± 0.12 ^d^	51.39 ± 0.48 ^abc^
	PFG:κC (1:2)	54.01 ± 0.18 ^d^	−3.77 ± 0.03 ^g^	−4.59 ± 0.07 ^d^	53.63 ± 0.19 ^ab^
7	FG:κC (1:2)	40.32 ± 0.66 ^i^	−1.31 ± 0.15 ^a^	−5.02 ± 0.14 ^g^	40.10 ± 0.67 ^c^
	PFG:κC (1:2)	44.56 ± 0.42 ^g^	−2.03 ± 0.12 ^c^	−6.25 ± 0.38 ^i^	44.24 ± 0.36 ^bc^
14	FG:κC (1:2)	41.21 ± 0.40 ^h^	−1.65 ± 0.06 ^b^	−4.73 ± 0.17 ^d^	40.99 ± 0.40 ^c^
	PFG:κC (1:2)	47.26 ± 0.22 ^f^	−2.35 ± 0.12 ^d^	−5.93 ± 0.21 ^h^	46.88 ± 0.23 ^bc^

The values are the mean ± standard deviation, and the values with different letters in the same column are significantly different at *p* < 0.05.

**Table 4 foods-12-03682-t004:** Changes in deformation resistance rate of JGs during storage.

Storage Time/Day	Sample Name	Deformation Resistance	Sample Name	Deformation Resistance
0 day	FG:κC (2:1)	0.93 ± 0.02 ^a^	PFG:κC (2:1)	0.94 ± 0.00 ^a^
	FG:κC (1:1)	0.78 ± 0.08 ^bc^	PFG:κC (1:1)	0.84 ± 0.02 ^b^
	FG:κC (1:2)	0.92 ± 0.02 ^a^	PFG:κC (1:2)	0.95 ± 0.02 ^a^
7 day	FG:κC (1:2)	0.67 ± 0.06 ^cd^	PFG:κC (1:2)	0.69 ± 0.05 ^d^
14 day	FG:κC (1:2)	0.45 ± 0.07 ^e^	PFG:κC (1:2)	0.44 ± 0.05 ^e^

The values are the mean ± standard deviation, and the values with different letters in the same column are significantly different at *p* < 0.05.

**Table 5 foods-12-03682-t005:** Statistics of bacterial/fungal sequencing results in JGs during storage.

Correlation Index	Species	Sample Name					
		FG:κC (1:2) 0 d	PFG:κC (1:2) 0 d	FG:κC (1:2) 7 d	PFG:κC (1:2) 7 d	FG:κC (1:2) 14 d	PFG:κC (1:2) 14 d
Original sequence number	Bacteria	131,359	120,967	131,101	127,243	120,750	125,948
	Fungus	90,878	84,545	85,608	86,426	88,285	87,826
OUT/pc	Bacteria	387	308	329	124	126	137
	Fungus	132	104	92	110	121	97
Kingdoms	Bacteria	1	1	1	1	1	1
	Fungus	3	3	3	4	4	3
Phyla	Bacteria	19	14	13	16	11	10
	Fungus	7	7	8	8	8	6
Classes	Bacteria	26	21	17	22	14	15
	Fungus	18	17	20	17	18	14
Orders	Bacteria	57	46	33	49	30	30
	Fungus	23	23	29	26	23	25
Families	Bacteria	104	89	56	91	50	54
	Fungus	30	30	30	31	33	32
Genera	Bacteria	162	145	69	134	60	75
	Fungus	25	24	25	24	28	27

**Table 6 foods-12-03682-t006:** Results of alpha diversity analysis of microbial communities in JGs during storage.

Correlation Index	Species	Sample Name					
		FG:κC (1:2) 0 d	PFG:κC (1:2) 0 d	FG:κC (1:2) 7 d	PFG:κC (1:2) 7 d	FG:κC (1:2) 14 d	PFG:κC (1:2) 14 d
Richness	Bacteria	387	308	329	124	126	137
	Fungus	132	104	92	110	121	97
Chao1	Bacteria	387.1	308.1	329.2	124.4	126.8	138.6
	Fungus	132.9	104.9	92.7	112.9	121.5	98.1
Reads	Bacteria	89,108	89,615	84,265	101,127	84,087	99,476
	Fungus	4394	3329	2169	3172	2204	4516
Simpson	Bacteria	0.179	0.36	0.166	0.497	0.387	0.522
	Fungus	0.276	0.0725	0.246	0.189	0.0869	0.293
Coverage	Bacteria	0.99	0.99	0.99	0.99	0.99	0.99
	Fungus	0.99	0.99	0.99	0.99	0.99	0.99

**Table 7 foods-12-03682-t007:** Bacterial composition and changes in JGs during storage.

Bacteria	Sample Name					
Genus	FG:κC (1:2) 0 d	PFG:κC (1:2) 0 d	FG:κC (1:2) 7 d	PFG:κC (1:2) 7 d	FG:κC (1:2) 14 d	PFG:κC (1:2) 14 d
Pseudomonas	4.01%	4.74%	13.35%	94.72%	90.87%	92.25%
Shewanella	34.03%	57.85%	32.21%	2.77%	4.51%	3.63%
Geobacillus	20.91%	13.42%	18.66%	0.84%	1.58%	1.27%
Anoxybacillus	8.79%	5.33%	7.84%	0.42%	0.70%	0.63%
Lactococcus	9.10%	4.26%	7.40%	0.25%	0.56%	0.63%
Acinetobacter	8.69%	4.48%	6.38%	0.20%	0.32%	0.28%
Thermus	4.30%	3.79%	5.60%	0.28%	0.65%	0.50%
Streptococcus	3.88%	1.77%	3.21%	0.12%	0.23%	0.26%
(Unassigned)	0.63%	0.55%	0.52%	0.16%	0.25%	0.18%
Bacillus	0.70%	0.55%	0.58%	0.03%	0.03%	0.04%
Chryseobacterium	0.66%	0.46%	0.56%	0.01%	0.02%	0.04%
Enhydrobacter	0.40%	0.23%	0.32%	0.01%	0.02%	0.02%
Carnobacterium	0.35%	0.24%	0.27%	0.01%	0.02%	0.03%
Aeromonas	0.33%	0.18%	0.28%	0.01%	0.01%	0.02%
Fusobacterium	0.21%	0.14%	0.23%	0.01%	0.01%	0.02%
Others	3.00%	2.00%	2.58%	0.14%	0.20%	0.21%

**Table 8 foods-12-03682-t008:** Fungal composition and changes in JGs during storage.

Fungus	Sample Name					
Genus	FG:κC (1:2) 0 d	PFG:κC (1:2) 0 d	FG:κC (1:2) 7 d	PFG:κC (1:2) 7 d	FG:κC (1:2) 14 d	PFG:κC (1:2) 14 d
(Unassigned)	79.29%	66.57%	78.24%	64.56%	64.20%	72.83%
Unidentified	10.70%	20.19%	13.14%	7.76%	11.12%	3.52%
Cutaneotrichosporon	2.28%	1.08%	1.94%	13.97%	8.12%	10.92%
Aspergillus	1.34%	4.84%	0.65%	3.66%	1.91%	3.21%
Kluyveromyces	1.00%	2.07%	1.24%	1.32%	4.67%	3.14%
Poaceascoma	2.41%	0.81%	0.69%	0.73%	0.50%	1.00%
Candida	0.30%	0.06%	0.14%	1.17%	1.09%	1.66%
Phaeosphaeria	0.09%	1.14%	0.37%	1.36%	0.54%	0.55%
Apiotrichum	0.11%	0.00%	0.00%	2.62%	0.77%	0.47%
Pyrenochaeta	0.09%	0.18%	0.60%	0.60%	1.00%	0.40%
Trichosporon	0.27%	0.72%	0.00%	0.19%	0.14%	0.11%
Cyberlindnera	0.02%	0.06%	0.51%	0.00%	0.36%	0.62%
Wallemia	0.48%	0.15%	0.18%	0.09%	0.68%	0.00%
Resinicium	0.00%	0.03%	0.00%	0.03%	1.77%	0.04%
Fusarium	0.41%	0.18%	0.09%	0.06%	0.27%	0.07%
Others	1.20%	1.91%	2.20%	1.88%	2.85%	1.45%

## Data Availability

The data presented in this article are available on reasonable request, from the corresponding author.

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
