# Peer review of "Phosphorylated Fish Gelatin and the Quality of Jelly Gels: Gelling and Microbiomics Analysis"

_foods, 2023, doi:10.3390/foods12193682_

Round 1

Reviewer 1 Report

I evaluated the manuscript entitled "The phosphorylated fish gelatin on the quality of jelly gels: Gelling and microbiomics analysis".
The article contains interesting information regarding the formulation and characterization.
Only some minor points: 
Clarify the novelty and contribution to knowledge of the manuscript. Add a schematic diagram to illustrate the used procedure in supplement. I would recommend a comprehensive comparison with the author's previous studies and others.

Reviewer 2 Report

The present study has a great practical approach by revealling improved gelation ability of phosphorylated fish gelatin. However several improvements must be brought

1 in the abstract state the purpose of the research and then in line 10 please clarify if it is only FG or also PFG usage in formulation. Then you should describe the methodology employed in the article not directly the sensory quality results. Describe the microbiology analysis methodology not only results.

2. In introduction in line 24 change swallowalbe.

add jelly manufacturing

in line 28 and 41 , 43 , 52 add citation

in line 67-69 mention the usage of k carragenan and motivate it

In line 88 mention the recipe origin

Line 106 provide cohesive energy formula

line 112 provide sample dimension

present sensory evaluation as 2.5.3 prior microbial diversity

in lines 151-152 you should consider the presence of skimmed milk proteins

In lines 154 is not clear if you discuss your results from this study that k carragenan increase improves the textural properties or you cite work of chen et all

Line 158, if no statistical diferences were recorded why did you choose further the jgs3 samples?in lines 171-172 complete with exact values and samples

Figure 2 should be modified. Include all pfg samples in one graph. Same for fg jellies.

table 1 should present the mean values and standard deviation

And statistical analysis

190-191 lines please take in regard the presence of the skimmed milk from the formulation

Line 200-201 include exact results and storage period.

line 204 correlate the results of microbiology with the textural degradation.

in line 207 include concrete results.

for the color analysis please search if there is any index wich measures the translucidity of the sample.

in lines 233 why do you mention day 0 if the protocol is only 24 hours (1 day)? Revisit methodology and complete the storage follow up.

Complete methodology with information related to alpha diversity index

Reviewer 3 Report

The manuscript, entitled "The Impact of Phosphorylated Fish Gelatin on the Quality of Jelly Gels: A Study in Gelling and Microbiomics Analysis," delves into the sensorial, textural, and microbiological aspects of jelly produced by combining fish gelatin with carrageenan. The subject matter is intriguing, and various facets of jelly gels have been thoroughly examined. Nevertheless, I recommend enhancing the abstract and introduction in terms of both language and grammar. I have already provided feedback on several language errors in the comments. Furthermore, I have suggested that the authors simplify the coding system for the treatments, as this research does not involve an extensive array of treatments. You can refer to my detailed comments below for more information:

Abstract

Line 9-10 more details about the background of the research and PFG is required.

Line 9: When authors acclaimed that PFG showed better physical and chemical properties than fish gelatin, do they mean in this study or previous research? Should clarify here.

Line 16-17: what is the difference between amount of M.O species and abundance of MO species? What is the microbial structure means?

Line 23: rephrase “leisure food for the public” please

Line 24: do you mean preparation procedure?

Line 27: “et al.” should be changed to “etc.”

Line 34: I suggest to say “diabetic patients” or “diabetics”

Line 34: remove “particular” please

Line 47-48: What is the reason that Muslim and Jews doesn’t eat food containing beef? As far as I know only Pork is forbidden in those religions and for muslims beef should be slaughtered in halal way

Line 81: preparation of jelly candies: The coding system is very confusing and hard to track while looking into graphs or table. I suggest as there is not many treatments used in this study, use an easier system to code the treatments such as PFG:kC (1:1), PFG:kC (1:2) and so on.

Did the author remove the sugar from the formulation of jelly gels? If so, how that effected the flavour acceptance of the samples?

Line 86: what essence did you use in this study?

Line 99: use the past tense form for evaluation. “evaluated”

Line 100: There is no “ tissue morphology” on the Table S2. I believe its “Organization status”. However, I suggest to use more standard sensorial terms for this section such as “textural acceptance”

Line 108: What type of texture analysis the author used in this study? Is it gel blooming, puncture, compression or another type of texture analysis?

Line 120. Please mention a reference for this method

Line 130: Was this the results from your study or that was the authors presumption? Also the sentence should be in past tense form

Line 141: use “statistical analysis” instead of “data processing”

Line 142: what does “SD” stands for? Do you mean the standard deviation? If so, it should be mentioned

Line 144: Did you draw pictures or used the software for generating graphs?

Line 155-156: The sentence needs to be rewritten

Figure 1: The main reason of using figures is to make the data easier to read. However, the spider web graph used in this study is confusing and I suggest the authors use these references to re-draw the graphs. The sensorial parameters should be on the graph and  

Khodaei, D., Hamidi-Esfahani, Z., & Rahmati, E. (2021). Effect of edible coatings on the shelf-life of fresh strawberries: A comparative study using TOPSIS-Shannon entropy method. NFS Journal23, 17-23.

Line 185-186: This claim can be incorrect as after a certain level of hardness and chewiness the acceptance of gels or any food product decreases.

Microbial analysis results: can you please compare the microbial count on the samples with the commercial jelly in the market or other literature reviews?

I recommend making moderate changes to the language of the manuscript, particularly in the abstract, introduction, and conclusion.
